

# Genetic assessment of farmed *Oreochromis mossambicus* populations in South Africa

Mahlatse Fortunate Mashaphu[1], Gordon Craig O'Brien[2], Colleen Thelma Downs[1] and Sandi Willows-Munro[1]

[1] Centre for Functional Biodiversity, School of Life Sciences, University of KwaZulu-Natal, Pietermaritzburg, KwaZulu-Natal, South Africa

[2] Gulbali Institute, Inland Fisheries Research Group, Charles Sturt University, Albury, New South Wales, Australia

## ABSTRACT

The global utilisation of *Oreochromis* spp. in freshwater aquaculture extends to South Africa. Here the native Mozambique tilapia (*Oreochromis mossambicus*) has been proposed as a priority species for regional aquaculture projects, although it is still not preferred over the non-native *O. niloticus*. There is limited understanding of the genetic diversity, and genetic differentiation of farmed *O. mossambicus* in South Africa. Using a suite of 14 microsatellite markers, the present study aimed to determine the origin and genetic diversity of four farmed *O. mossambicus* populations in KwaZulu-Natal and Mpumalanga provinces. Wild *O. mossambicus* from rivers surrounding the farms were included to trace the origin of farmed populations. Results revealed lower genetic diversity in farmed populations compared to wild populations. In particular, the University of Zululand population exhibited lower genetic diversity compared to the rest of the farmed populations. While most farmed populations closely resembled their local wild counterparts, the uMphafa ponds exhibited distinct genetic characteristics. Notably, some individuals from uMphafa shared genetic affinities with those from the Thukela River, suggesting that the Thukela River could be the source of this farmed population, or that farmed fish may have been introduced or escaped into the river. The study suggests that select farmed populations may have the potential for use in breeding and broodstock supplementation programs but emphasizes the importance of thorough genetic monitoring. However, before these populations can be considered for broodstock supplementation, further investigation is required to confirm their genetic integrity and rule out potential contamination from invasive species.

# INTRODUCTION

Anthropogenic activities have a major impact on both wild and farmed fish stocks, and freshwater aquaculture practices have led to changes in gene flow patterns of wild fish populations and the release of invasive species into native ecosystems (*Svåsand et al., 2007*; *Crispo et al., 2011*; *Firmat et al., 2013*; *Garg et al., 2014*; *Bolstad et al., 2021*). One of the key genetic risks associated with aquaculture is hybridisation between farmed and wild stocks, which can lead to the introgression of non-native genes into wild populations, disrupting local adaptations and decreasing the genetic integrity of native species (*Glover et al., 2010*;

Corresponding author
Sandi Willows-Munro,
willows-munro@ukzn.ac.za

*Firmat et al., 2013*; *May et al., 2024*). In particular, the release or escape of farmed tilapia can exacerbate hybridisation and genetic homogenisation, especially in areas where non-native species are introduced for aquaculture (*Romana-Eguia et al., 2004*; *Kang et al., 2023*; *May et al., 2024*). Baseline data on the genetic diversity and population structure of both farmed and wild populations of species used in aquaculture are essential for the sustainable management of farmed stocks and for ensuring the conservation and genetic health of wild populations over time (*Cossu et al., 2021*; *Sonesson et al., 2023*). Wild populations serve as an important genetic reservoir for aquaculture, providing essential genetic diversity that can enhance the resilience and adaptability of farmed stocks (*Brummett, 2008*; *Lind, Brummett & Ponzoni, 2012*; *Garg et al., 2014*; *Aguiar et al., 2018*; *Sonesson et al., 2023*). Monitoring changes in genetic variation in farmed populations is crucial because inbreeding and genetic drift in small, farmed populations can lead to inbreeding depression, reducing reproductive fitness and survivorship, ultimately lowering productivity (*Glover et al., 2010*; *Khadher et al., 2016*; *Aguiar et al., 2018*; *Lal et al., 2021*).

Members of the *Oreochromis* genus, particularly Nile tilapia (*O. niloticus*), Mozambique tilapia (*O. mossambicus*), and blue tilapia (*O. aureus*), are widely used in aquaculture due to their hardiness, ease of cultivation, and rapid growth (*FAO, 2002*; *FAO, 2019*; *Arumugam et al., 2023*). However, poor management of tilapia genetic resources, both in the wild and in culture, has posed significant challenges (*Kocher et al., 1998*; *Nobile et al., 2020*; *Gozlan et al., 2024*). Hybridisation between *O. niloticus* and with native *O. mossambicus*, especially in South Africa, threatens the genetic integrity and long-term viability of wild *O. mossambicus* populations, which are classified as Vulnerable by the IUCN (*Bills, 2019*). This hybridisation compromises important traits of *O. mossambicus*, such as tolerance to drought, high salinity, and low temperatures, which are vital for survival in its natural habitat (*Moralee, Van Der Bank & Van Der Waal, 2000*; *D'Amato et al., 2007*; *Geletu & Zhao, 2023*). Additionally, *O. niloticus* competes with *O. mossambicus* for food, territory, and breeding sites, exacerbating the pressures on native populations (*Esterhuyse, 2002*; *Geletu & Zhao, 2023*). To safeguard these genetic resources, improved management practices and stricter regulations are needed, including better protection of wild stocks and more rigorous enforcement of policies in farmed tilapia populations.

Based on estimates by the Tilapia Association of South Africa, there may be as many as 74 producers of tilapia in South Africa (*DAFF, 2016*). None of these facilities implement routine genetic monitoring of their farmed stock. This study evaluated the genetic diversity and differentiation within four farmed *O. mossambicus* localities: three aquaculture facilities from KwaZulu-Natal (Zini Fish Farm, University of Zululand Fish Farm and uMphafa ponds) and one (Fresca Fisheries Farm) from Mpumalanga, South Africa. In the KwaZulu-Natal and Mpumalanga provinces of South Africa, *O. mossambicus* occurs in all the major river catchments and is also farmed. Aquaculture practices suggested that the founder stock for farmed stocks comes from rivers surrounding the fish farms (various pers comm.). There is potential, however, for translocations to have occurred, with farms stocked using fish from many different river catchments. If these fish are released into natural systems, this could disrupt existing patterns of genetic variation in wild populations, leading to genetic homogenisation, reduced resilience, and adaptability

of natural populations (*Pinter, Epifanio & Unfer, 2019*; *May et al., 2024*). Studies on farmed and wild tilapia in other regions have demonstrated that translocations between water bodies and fish farms can introduce genetic contamination, leading to the erosion of local genetic diversity and increased vulnerability of wild stocks to environmental changes (*Glover et al., 2010*; *Aguiar et al., 2018*; *Tibihika, Meimberg & Curto, 2022*; *Sonesson et al., 2023*). This highlights the need for comprehensive genetic monitoring of farmed stocks to prevent genetic homogenisation and to safeguard wild populations from hybridisation and inbreeding depression (*Khadher et al., 2016*; *Lal et al., 2021*; *Sonesson et al., 2023*). To determine the genetic admixture of the farmed individuals, wild *O. mossambicus* from five rivers in KwaZulu-Natal (Mfolozi, Mhlathuze, Matigulu, Thukela, and Mvoti) and three potential sources of fish in Mpumalanga (Komati River, Loskop Dam and Pieter Vorster Dam) surrounding the aquaculture farms were included. While a previous study (*Mashaphu et al., 2024*) focused on the genetic diversity and population structure of wild *Oreochromis mossambicus* populations across South Africa, this study evaluates farmed populations. Specifically, it examines their genetic diversity and relationships with local wild populations, and the potential for these farmed stocks to serve as broodstock for aquaculture in KwaZulu-Natal and Mpumalanga. This information is crucial for informing conservation strategies and improving management practices in aquaculture to ensure the long-term sustainability and genetic health of these populations.

## MATERIALS & METHODS

### Study area

The study was conducted in the KwaZulu-Natal and Mpumalanga provinces of South Africa (Fig. 1). KwaZulu-Natal, being a coastal province, presents opportunities for both marine and freshwater aquaculture because of its warm climate (*DAFF, 2016*; *Oyeleke, 2017*). Similarly, Mpumalanga, with its favourable climate, is conducive to freshwater aquaculture (*Oyeleke, 2017*). In the KwaZulu-Natal Province, three localities engage in the cultivation of Mozambique tilapia. These farms include Zini Fish Farm, which is connected to the Mlalazi estuary, the University of Zululand Aquaculture Research Unit, and the uMphafa Private Nature Reserve, which utilises earthen ponds for Mozambique tilapia breeding. The *O. mossambicus* at Zini Fish Farm are thought (Gordon O'Brien, pers. comm., 2018) to have originated from the Mlalazi estuary and surrounding northern KwaZulu-Natal rivers (Mfolozi, Matigulu, Mhlathuze, Thukela, and Mvoti). The establishment of *O. mossambicus* at Zini Fish Farm resulted from their thriving in earthen ponds initially designated for dusky kob (*Argyrosomus japonicus*) breeding. The University of Zululand is likely originated from fish collected in northern KwaZulu-Natal (Gordon O'Brien, pers. comm., 2018), although the exact source remains unclear. The broodstock source for the uMphafa ponds is also uncertain. However, considering the proximity of these ponds to the Thukela River in northern KwaZulu-Natal, individuals from the Thukela River were included in the study to investigate if it could be the source of the uMphafa ponds locality. In Mpumalanga, one farm (Fresca Fisheries Farm), presently cultivates *O. mossambicus*

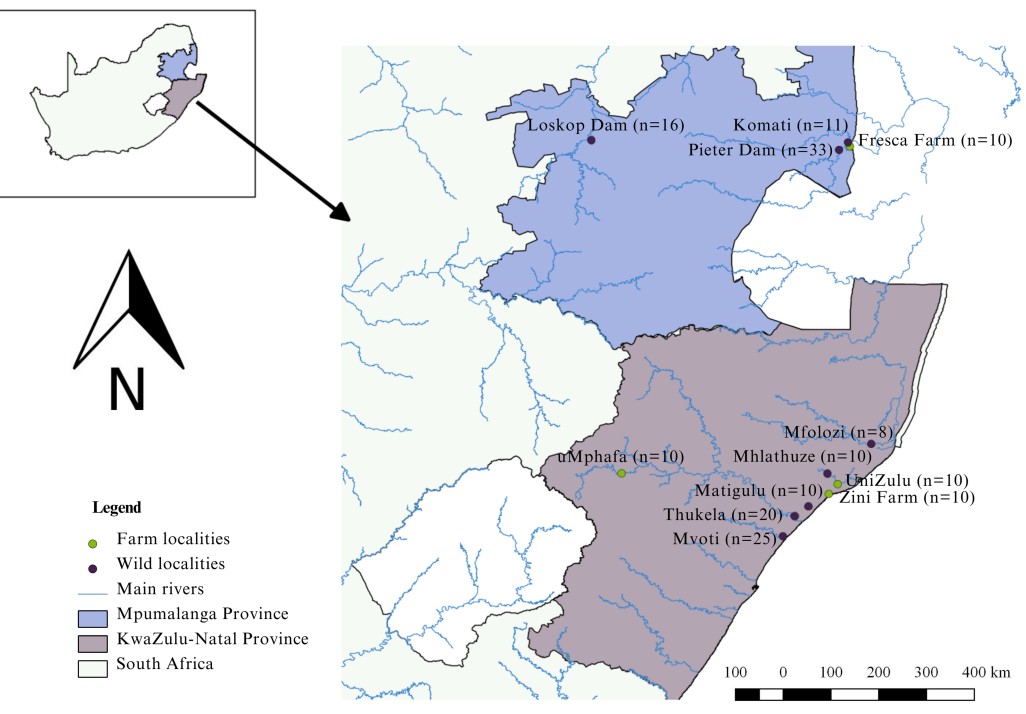

**Figure 1** Sampling localities of farmed and wild-caught *O. mossambicus* in KwaZulu-Natal and Mpumalanga, South Africa.

believed (Holt Lance, pers. comm., 2021) to have originated from the Komati River and the Loskop Dam, which are linked to the Olifants River catchment.

## Collection of DNA samples

The study adhered to all necessary ethical and legal protocols. Permits for sampling were diligently obtained from Ezemvelo KwaZulu-Natal Wildlife (OP1583/2017, OP1432/2018, and OP871/2021) and Mpumalanga Tourism and Parks Agency (MPB. 56932). Moreover, the study received ethical clearance from the University of KwaZulu-Natal Animal Ethics Subcommittee (REF: AREC/023/020). DNA sampling involved the collection of a total of 173 fin clips from both farmed and wild populations. Specifically, ten samples were collected from each of the four aquaculture farms: Zini Fish Farm, University of Zululand Aquaculture Facility, uMphafa ponds, and Fresca Fisheries (Fig. 1), contributing 40 farmed samples. Additionally, 133 fin clips were obtained from wild populations in the surrounding river catchments of these farms during 2017–2018. In KwaZulu-Natal, wild samples were sourced from the Mfolozi ($n = 8$), Mhlathuze ($n = 10$), Matigulu ($n = 10$), Thukela ($n = 20$), and Mvoti ($n = 25$) rivers. In Mpumalanga, wild individuals were collected from the Komati River ($n = 11$), Pieter Vorster Dam ($n = 33$), and Loskop Dam ($n = 16$) (Fig. 1).

Collection methods varied based on the habitat. Fish from farms were obtained from ponds and tanks using a scooping net and seine net. Wild specimens, on the other hand, were collected using established passive techniques, including fyke nets, electro-fishing, seine nets, and gill nets. Upon capture, *Oreochromis mossambicus* individuals were identified

to species level following *Skelton (2001)* and anesthetised using clove oil (0.5 ml/L) as per *Bennett et al. (2016)*. Morphometric measurements, including weight (g), standard length (mm), and total length (mm), were recorded for each fish. Additionally, each individual was photographed for reference. A non-lethal fin clip, approximately five $mm^2$ to 10 $mm^2$, was taken from each anesthetised fish, promptly preserved in 99% ethanol, stored in a refrigerator, and transported to the laboratories of the University of KwaZulu-Natal for subsequent genetic analyses. The captured fish were then placed in a recovery bucket and released back into their respective capture locations.

## DNA extraction and amplification

The DNA extraction from fin clips was performed using the Nucleospin® Tissue kit (Macherey-Nagel, Düren, Germany) following the manufacturer's standard protocol for animal tissue. Approximately 25 mg of fin tissue was finely cut and lysed using proteinase K. The resulting DNA was bound to silica membranes within spin columns, washed to eliminate impurities, and finally eluted in 100 µL of buffer, yielding purified DNA for further analysis. Extracted DNA samples were then stored at −20 °C. This study utilised 14 microsatellite loci, which were selected based on their prior successful application in previous studies (*Saju, Lee & Orban, 2010*; *Simbine, Viana de Silva & Hilsdorf, 2014*; *Bbole et al., 2020*; *Thomas et al., 2021*; *Mashaphu et al., 2024*). The loci were originally isolated from *O. niloticus* (*Hall, 2001*; *Simbine, Viana de Silva & Hilsdorf, 2014*) and *O. mossambicus* (*Saju, Lee & Orban, 2010*), making them well-suited for genetic analysis in the context of tilapia species. These markers have been demonstrated to be effective in assessing genetic diversity and population structure, as evidenced by their use in other studies involving *O. mossambicus* (*Mashaphu et al., 2024*) and related species (Table 1). The choice of these loci aligns with the objectives of this study to ensure robust and comparable results within the broader framework of tilapia genetic research. In each primer pair, the forward primer was fluorescently labelled. The PCR profiles were initially conducted as singleplexes for each of the 14 microsatellites. Subsequently, working primers were selected and grouped into multiplexes (A, B, C, and D) based on annealing temperature, fluorescence dye, and product size. Multiplex A and D had an annealing temperature of 60 °C, while multiplex B and C had an annealing temperature of 61 °C. Each PCR reaction was prepared to a final volume of 10 µl, comprising 0.1 µl of each primer, 5 µl of 2G Fast Multiplex Mix (KAPA Biosystems, Cape Town, South Africa), 0.3 µl of BSA, 4.2 µl of distilled water, and 0.3 µl of DNA. Thermocycler conditions included an initial 3 min at 94 °C, followed by 35 cycles of denaturation at 94 °C for 15 s, annealing for 30 s, extension at 72 °C for 1 min, and a final extension step at 72 °C for 10 min. The annealing temperature was set at 60 °C for multiplex A and D, and 61 °C for multiplex B and C. Fragment analyses were conducted at the Central Analytical Facilities (CAF), University of Stellenbosch, South Africa, using a 3500XL Genetic Analyzer from Applied Biosystems (Thermo Fisher). To ensure genotype accuracy, 20% of all individuals underwent re-amplification (https://zenodo.org/doi/10.5281/zenodo.11085961).
**Table 1** Microsatellite loci used in the present study for genotyping of *O. mossambicus* farmed and wild populations. Fluorescent dye and annealing temperature used in the present study are provided.

| Multiplex loci | Locus | Sequence | Fluorescent dye | Annealing temperature °C | GenBank ID |
|---|---|---|---|---|---|
| A | OM01[c] | F: TTTAAAGTTACACAGCAGTACAAAG<br>R: TTGTAGCATTTCAACACAGTCTC | Fam | 60 | GU391020 |
| D | OM02[c] | F: TGTGAATTTGACAACTTCCTTTC<br>R: ATCCTTGCAATAAGGTTACAG | Fam | 60 | GU391021 |
| D | OM03[c] | F: CTTTTTAATGAGCAACTTTTAAGTC<br>R: TGTGAATTTGACAACTTCCTTTC | Hex | 60 | GU391022 |
| A | OM04[c] | F: AGCTCAAAACCTCATACAAAGG<br>R: GCAGAGATGTCAGATGTTGTTC | Fam | 60 | GU391023 |
| B | OM05[c] | F: GTAAAGTTTGGAACAGAAATGCT<br>R: GATCACTTTTGGACAGACTGG | Hex | 61 | GU391024 |
| B | OM06[c] | F: TGAGCTACCGTAAGGATGTAC<br>R: GTTATTTCAATTATATTTGCATG | Fam | 61 | GU391025 |
| C | OM07[c] | F: TTGGCTCAGAGTGGTCAGG<br>R: CGCGTGGACTAAAAGCCAG | Hex | 61 | GU391026 |
| B | OM08[c] | F: TGTTGGTTGGATTACTGGG<br>R: GCTGTAATGGTTTTGAGGC | Fam | 61 | GU391027 |
| B | OM09[c] | F: GGCTACAACACCTGGATGG<br>R: TTGGGCTTACTGAAGCTGAC | Hex | 61 | GU391028 |
| C | UNH104[a] | F: GCAGTTATTTGTGGTCACTA<br>R: GGTATATGTCTAACTGAAATC | Tet | 61 | G12257 |
| C | UNH129[a] | F: AGAAGTCGTGCATCTCTC<br>R: TGTACATCATCTGTGGG | Tet | 61 | G12282 |
| B | UNH142[b] | F: CTTTACGTTGACGCAGT<br>R: GTGACATGCAGCAGATA | Tet | 61 | G12294 |
| B | UNH222[b] | F: CTCTAGCACACGTGCAT<br>R: TAACAGGTGGGAACTCA | Tet | 61 | G12373 |
| C | UNH231[b] | F: GCCTATTATAGTCAAAGCGT<br>R: ATTTCTGCAAAAGTTTTCC | Tet | 61 | G12382 |

**Notes.**

Primers taken from *Hall (2001)*[a] *Simbine, Viana de Silva & Hilsdorf (2014)*[b] and *Saju, Lee & Orban (2010)*[c].

## Molecular data analyses

Genotypes were scored using GeneMarker® version 2.4 (*Hulce et al., 2011*). Null allele frequencies were computed for both farmed and wild *O. mossambicus* individuals using FreeNA version 20091116 (*Chapuis & Estoup, 2006*). To assess whether null alleles significantly influenced population structure estimates, both null allele-excluded (ENA) corrected and non-corrected global $F_{ST}$-values were calculated in FreeNA. Subsequently, these values were compared using a student $t$-test in Microsoft Excel 2016.

## Genetic diversity

Polymorphic information content (PIC) for each microsatellite locus was computed using Cervus version 3.0.7 (*Marshall et al., 1998*; *Kalinowski, Taper & Marshall, 2007*). PIC serves as a metric indicating the suitability of each microsatellite locus for assessing genetic differentiation between farmed and wild sampling localities (*Botstein et al., 1980*). Loci with PIC > 0.5 are highly informative, while those with PIC < 0.25 are considered less

informative (*Botstein et al., 1980*). The total number of alleles (A), effective number of alleles (Ae), unbiased expected and observed heterozygosity (uH$_E$ and H$_O$) for the 14 microsatellites amplified from both farmed and wild *O. mossambicus* were calculated using GenAlEx version 6.5 (*Peakall & Smouse, 2012*). Allelic richness values were obtained using FSTAT version 2.9.3.2 (*Goudet, 2001*). Tests for deviation from Hardy–Weinberg equilibrium (HWE) for markers and populations were performed using GenAlEx. Linkage disequilibrium (LD) was estimated using Genepop version 4.7.0 (*Raymond & Rousset, 1995*; *Rousset, 2008*). To assess the significance of LD between pairs of loci, chi-square *p*-values were obtained using the Markov Chain method with 10,000 replicates, 100 groups, and 5,000 iterations per group. This approach was used to determine whether non-random associations existed between loci, which can provide insights into population structure. Bonferroni corrections were applied to both the HWE and LD tests to account for multiple comparisons and ensure the robustness of the statistical analyses. The inbreeding coefficient (Fis) was calculated using GenAlEx. Positive high Fis-values indicate high levels of inbreeding, characterised by an excess of homozygotes, while negative Fis-values indicate outbreeding and an excess of heterozygosity in the population (*Simbine, Viana de Silva & Hilsdorf, 2014*).

## Population genetic structure

The Bayesian clustering method, implemented in STRUCTURE version 2.3.4 (*Pritchard, Stephens & Donelly, 2000*), was employed to identify potential admixture within the farmed and wild *O. mossambicus* sampling localities from KwaZulu-Natal and Mpumalanga. Assignment tests were conducted using the admixture model with correlated allele frequencies. The STRUCTURE analyses comprised 500,000 Markov-Chain Monte Carlo (MCMC) replicates with a 50,000 generation burn-in period. Ten iterations were performed for K-values ranging from 1 to 10. The STRUCTURE selector software version 2.3 (*Li & Liu, 2018*) was utilised to determine the optimal number of genetic clusters (K-value) using the Puechmaille method (*Puechmaille, 2016*). STRUCTURE harvester version 0.6.94 (*Earl & VonHoldt, 2012*) was employed to obtain membership coefficient values (*Q*-values) for the optimal K. Bar plots for optimal K-values were generated using Clumpak version 1.1 (*Kopelman et al., 2015*). Population structure was visualised by conducting Principal Coordinate Analysis (PCoA) based on the genetic distance matrix calculated using Nei's genetic distances (*Nei, 1972*) in GenAlEx. This approach allowed us to represent the genetic relationships among individuals and groups in a multidimensional space, highlighting the similarities and differences in their genetic profiles.

Analysis of Molecular Variance (AMOVA) in GenAlEx was employed to assess the significance of genetic differentiation among all sites (*Excoffier, Smouse & Quattro, 1992*). The test, based on 9,999 permutations, first evaluated the significance of genetic differentiation between farmed and wild sampling localities from KwaZulu-Natal and Mpumalanga. Pairwise $F_{ST}$-values, quantifying genetic differentiation between farmed and wild *O. mossambicus* using the ENA correction method in FreeNA, were calculated.The significance of genetic differentiation between farmed and wild *O. mossambicus* was assessed using FSTAT software (version 2.9.3.2; *Goudet, 2001*). We calculated *p*-values
**Table 2  Genetic diversity indices of the 14 microsatellite loci of the four farmed and eight wild *O. mossambicus* populations in the present study.**

| Locus | A | Null allele frequencies | $F_{ST}{}^{A}$ | $F_{ST}{}^{B}$ | Ar | $H_O$ | $uH_E$ | $F_{is}$ | PIC |
|---|---|---|---|---|---|---|---|---|---|
| OM01 | 8 | 0.11 | 0.15 | 0.13 | 6.06 | 0.57 | 0.80 | 0.26 | 0.87 |
| OM02 | 5 | 0.14 | 0.21 | 0.21 | 4.17 | 0.29 | 0.54 | 0.45 | 0.82 |
| OM03 | 5 | 0.13 | 0.22 | 0.22 | 4.22 | 0.33 | 0.55 | 0.37 | 0.85 |
| OM04 | 8 | 0.04 | 0.09 | 0.09 | 6.43 | 0.82 | 0.83 | −0.04 | 0.88 |
| OM05 | 6 | 0.15 | 0.20 | 0.16 | 4.96 | 0.51 | 0.75 | 0.28 | 0.90 |
| OM06 | 2 | 0.13 | 0.52 | 0.49 | 1.94 | 0.16 | 0.30 | 0.43 | 0.68 |
| OM07 | 9 | 0.02 | 0.09 | 0.09 | 7.04 | 0.92 | 0.84 | −0.14 | 0.90 |
| OM08 | 3 | 0.08 | 0.27 | 0.27 | 2.65 | 0.43 | 0.48 | 0.07 | 0.73 |
| OM09 | 5 | 0.08 | 0.30 | 0.30 | 3.83 | 0.45 | 0.56 | 0.15 | 0.77 |
| UNH104 | 5 | 0.17 | 0.21 | 0.18 | 3.90 | 0.38 | 0.65 | 0.38 | 0.78 |
| UNH142 | 3 | 0.18 | 0.30 | 0.25 | 2.61 | 0.20 | 0.45 | 0.53 | 0.75 |
| UNH129 | 5 | 0.07 | 0.13 | 0.13 | 3.56 | 0.42 | 0.48 | 0.09 | 0.69 |
| UNH222 | 3 | 0.25 | 0.25 | 0.22 | 3.08 | 0.13 | 0.53 | 0.74 | 0.75 |
| UNH231 | 2 | 0.09 | 0.10 | 0.18 | 2.08 | 0.10 | 0.22 | 0.55 | 0.50 |
| **Mean** | 5 | 0.12 | 0.22 | 0.21 | 4.04 | 0.41 | 0.57 | 0.30 | 0.78 |

**Notes.**
A, number of alleles; Ar, Allelic richness; $F_{ST}^{A}$, non-corrected; $F_{ST}$, $F_{ST}^{B}$, ENA corrected $F_{ST}$; $H_O$, observed heterozygosity; $uH_E$, unbiased expected heterozygosity; $F_{is}$, Inbreeding coefficient; PIC, polymorphic information content.

based on 1,320 permutations to assess the significance of $F_{ST}$-values, which measure genetic differentiation between populations. A significance level of 0.000758 was adopted after the Bonferroni correction to control for the error rate.

## RESULTS

A total of 173 individuals from four farmed and eight wild *O. mossambicus* from KwaZulu-Natal and Mpumalanga were successfully genotyped using 14 microsatellite loci. Not all microsatellites were amplifiable in all individuals, and some missing data were included in the data matrix, but no microsatellite locus had more than 27% missing data. Fourteen loci were highly polymorphic for the farmed and wild *O. mossambicus,* with PIC values ranging from 0.50 (UNH231) to 0.90 (OM05 and OM07) (Table 2). Null allele frequencies of the 14 loci varied from 0.02 (OM07) to 0.25 (UNH222) (Table 2). Both corrected (excluding null alleles, ENA) and non-corrected $F_{ST}$ values were computed and there were no discernible differences between the $F_{ST}$ values, indicating that the inclusion of null alleles did not influence the estimation of population structure. All data, including those with null alleles, were retained for subsequent analyses.

### Genetic diversity
The microsatellite loci employed in this study exhibited a mean of five alleles per locus. The number of alleles ranged from two (OM06) to nine (OM07), demonstrating substantial variability across the markers (Table 2). Allelic richness, a measure accounting for sample size differences, varied from 1.94 (OM06) to 7.04 (OM07). The observed heterozygosity
varied significantly, ranging from 0.10 (UNH231) to 0.92 (OM07). Notably, the unbiased expected heterozygosity was consistently higher than the observed heterozygosity across all loci, suggesting potential inbreeding within these populations, as reflected in the $F_{is}$-values. The unbiased expected heterozygosity ranged from 0.22 (UNH231) to 0.84 (OM07) (Table 2). While most loci exhibited relatively low inbreeding coefficients, exceptions such as UNH142, UNH222, UNH231, and OM02 showed Fis values approaching 1, indicating a pronounced deficiency of heterozygotes. The spectrum of $F_{is}$-values ranged from −0.04 (OM04) to 0.74 (UNH222) (Table 2).

The farmed *O. mossambicus* sampling localities exhibited a mean recovery of 4 alleles, whereas the wild sampling localities displayed a higher mean of 6 alleles. The number of alleles observed in both farmed and wild sampling localities varied across locations, ranging from 3 (uMphafa ponds) to 8 (Pieter Vorster Dam) (Table 3). The effective number of alleles for farmed sampling localities ranged from two (uMphafa ponds) to three (Zini Fish Farm), while wild sampling localities showed variability from three (Matigulu) to five (Loskop Dam). The mean observed heterozygosity (Ho) in farmed populations was 0.43, while in wild populations it was 0.40 (Table 3). Across both farmed and wild populations, observed heterozygosity was generally lower than the unbiased expected heterozygosity ($uH_E$), with a standard deviation of 0.11. An exception was observed in the uMphafa ponds, where the $H_O$ exceeded $uH_E$ (Table 3). The mean $uH_E$ was 0.53 for farmed populations and 0.59 for wild populations. Observed heterozygosity ranged from 0.31 to 0.51, while $uH_E$ varied from 0.46 to 0.73 across all populations (Table 3). These values highlight the genetic variation within the sampled populations and are reflective of the genetic diversity observed across both farmed and wild populations in this study.The mean $F_{is}$-value was lower in farmed sampling localities (0.17) than in wild sampling localities (0.35), with values ranging from −0.10 in the uMphafa ponds to 0.45 in Pieter Vorster Dam (Table 3). No significant deviations from HWE were observed in any farmed or wild sampling localities except for the Pieter Vorster Dam. This finding is consistent with the high $F_{is}$ value reported for this sampling locality. Based on the genetic diversity analyses, the Zini Fish Farm exhibited higher genetic diversity followed by the Fresca Fisheries Farm compared with the University of Zululand and uMphafa ponds sampling localities. Among the eight wild sampling localities, the Mhlathuze, Komati, and Loskop Dam sampling localities displayed higher genetic diversity than the other wild sampling localities. The overall genetic diversity was lower in farmed sampling localities compared to wild sampling localities.

## Population structure and genetic differentiation

The STRUCTURE analyses conducted on both farmed and wild sampling localities, utilising the admixture model, identified $K = 9$ as the optimal sub-population strategy, as determined by the Puechmaille method. Additionally, $K = 8$ and $K = 10$ were presented for comparative purposes (Fig. 2). The overall STRUCTURE clustering pattern revealed that the farmed *O. mossambicus* populations exhibited significant genetic connections with their wild counterparts from adjacent river systems. Specifically, the Zini Fish Farm and the University of Zululand populations displayed notable genetic similarities with wild

**Table 3** Genetic diversity indices of the eight wild and four farmed populations of *O. mossambicus* from KwaZulu-Natal and Mpumalanga.

| Sampling locality | N | A | Ae | Ar | $H_O$ | $uH_E$ | $F_{is}$ |
|---|---|---|---|---|---|---|---|
| *Farmed populations* | | | | | | | |
| Zini farm | 10 | 5 | 3 | 2.74 | 0.38 | 0.59 | 0.31 |
| UniZulu ponds | 10 | 4 | 3 | 2.50 | 0.36 | 0.48 | 0.19 |
| uMphafa ponds | 10 | 3 | 2 | 2.01 | 0.51 | 0.46 | −0.10 |
| Fresca Farm | 10 | 4 | 3 | 2.28 | 0.47 | 0.58 | 0.28 |
| **Mean** | 10 | 4 | 3 | 2.38 | 0.43 | 0.53 | 0.17 |
| *Wild populations* | | | | | | | |
| Mfolozi | 8 | 4 | 3 | 1.97 | 0.31 | 0.51 | 0.37 |
| Mhlathuze | 10 | 5 | 3 | 2.44 | 0.46 | 0.57 | 0.20 |
| Matigulu | 10 | 3 | 3 | 2.37 | 0.36 | 0.46 | 0.32 |
| Thukela | 19 | 6 | 4 | 3.35 | 0.37 | 0.58 | 0.37 |
| Mvoti | 25 | 6 | 3 | 3.90 | 0.39 | 0.54 | 0.26 |
| Komati | 10 | 6 | 4 | 2.42 | 0.45 | 0.73 | 0.38 |
| Pieter Dam | 32 | 8 | 3 | 4.88 | 0.37 | 0.65 | 0.45 |
| Loskop Dam | 16 | 7 | 5 | 3.65 | 0.45 | 0.69 | 0.41 |
| **Mean** | 16 | 6 | 3 | 3.12 | 0.40 | 0.59 | 0.35 |

Notes.

N, number of individuals; A, number of alleles; Ae, effective number of alleles; Ar, Allelic richness; $uH_E$, unbiased expected heterozygosity; $H_O$, observed heterozygosity; $F_{is}$, inbreeding coefficient are provided for each population.

*O. mossambicus* from the Mfolozi, Mhlathuze, and Matigulu rivers (Fig. 2). Additionally, some genetic affinities were observed between the Zini Fish Farm and wild individuals from the Mvoti and Thukela rivers. In contrast, the farmed populations from the uMphafa ponds exhibited unique genetic patterns, suggesting a founding lineage from individuals sourced from the Thukela River. Notably, the uMphafa ponds also shared genetic components with individuals from the Loskop Dam and the Komati River. The Fresca Fisheries Farm displayed distinct genetic characteristics, with admixture components primarily reflecting those found in farmed populations across KwaZulu-Natal, indicating the potential movement of broodstock among these sites. Conversely, the wild *O. mossambicus* from Pieter Vorster Dam appeared to be highly genetically distinct, forming an exclusive genetic cluster (Fig. 2).

The principal coordinate analysis (PCoA) further highlighted the genetic relationships among the farmed and wild *O. mossambicus* populations. Specifically, the farmed *O. mossambicus* from Zini Fish Farm and the University of Zululand clustered closely, indicating potential shared source populations (Fig. 3A). The genetic proximity of these farmed individuals to wild *O. mossambicus* from the Matigulu, Mfolozi, and Mhlathuze rivers suggested a degree of genetic similarity (Fig. 3A). Consistent with the STRUCTURE results, the uMphafa ponds appeared isolated and did not cluster distinctly with any of the wild *O. mossambicus* populations (Fig. 3). However, some individuals from the Thukela River shared genetic affinities with those from the uMphafa ponds (Fig. 3B), suggesting that the Thukela River could be the source of the uMphafa ponds' broodstock, or that some individuals from the uMphafa farm may have escaped into the Thukela River. The
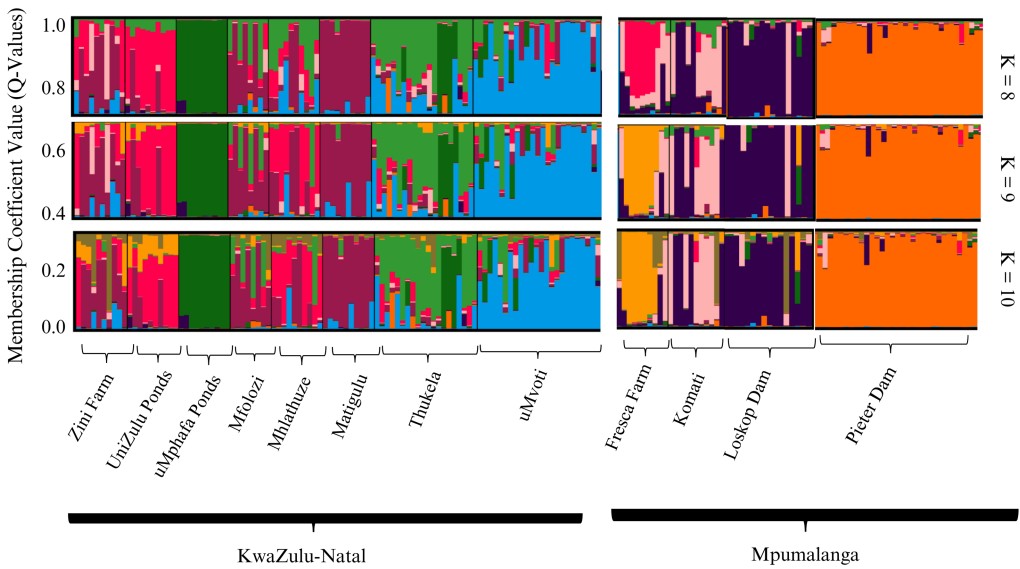

**Figure 2  STRUCTURE bar plots of farmed and wild *O. mossambicus* populations in KwaZulu-Natal and Mpumalanga based on coefficient values (Q-values). The most likely recovered optimal value of genetic clusters was *K* = 9.**

Fresca Fisheries Farm clustered with the wild *O. mossambicus* from the Komati River and Loskop Dam but did not show a genetic relationship with the *O. mossambicus* from Pieter Vorster Dam (Fig. 3).

The Nei's unbiased genetic distances supported the observed PCoA clustering patterns, with distances ranging from 0.06 between the University of Zululand and Mhlathuze River to 0.98 between uMphafa and Mhlathuze River (Table S1). Generally, the farmed sampling localities displayed genetic similarity to the surrounding wild *O. mossambicus*, except for the uMphafa ponds, which exhibited genetic distinctiveness from the nearby wild populations in KwaZulu-Natal. However, these particular farmed individuals showed limited genetic similarities with wild *O. mossambicus* from the Thukela River, corroborating the STRUCTURE results. The shared alleles between the farmed *O. mossambicus* (Zini Fish Farm and University of Zululand) suggested a common source for their establishment. The Fresca Fisheries Farm demonstrated genetic similarity to the wild *O. mossambicus* from the Komati River and Loskop Dam, supporting these as potential sources for this farm (Table S1). Based on Nei's unbiased genetic distances, the *O. mossambicus* from the uMphafa ponds also clustered more closely with farmed *O. mossambicus* from Fresca Fisheries and wild *O. mossambicus* from the Komati River and Loskop Dam (Table S1). This clustering suggests potential movement or exchange of broodstock between the two provinces.

The AMOVA results revealed significant genetic differentiation between the farmed and wild *O. mossambicus*. The genetic variations observed among the farmed and wild *O. mossambicus* accounted for 21% among groups (farmed *vs.* wild), 27% among sites within groups, and 52% within sites (Table 4). Pairwise $F_{ST}$ comparisons between various

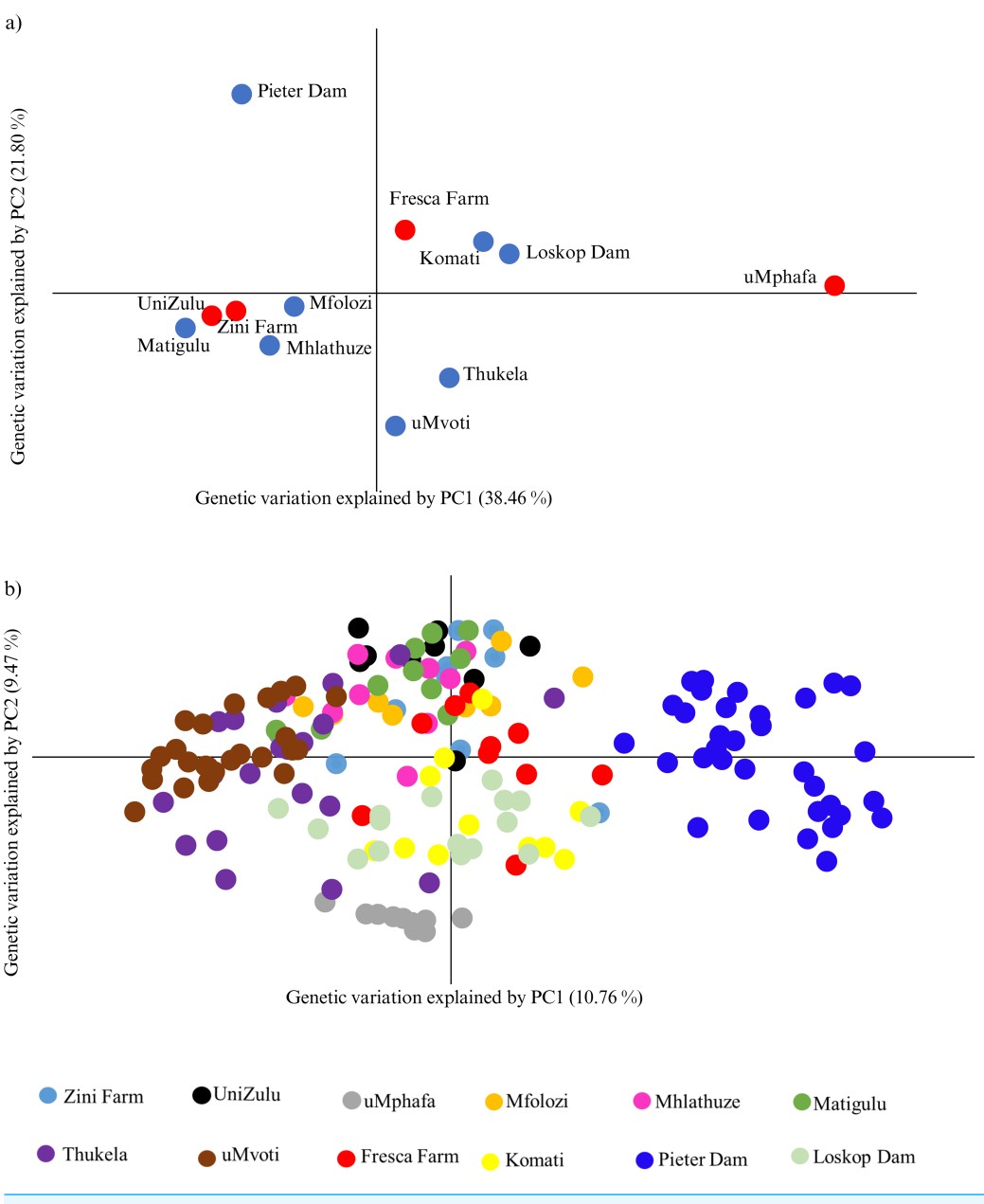

**Figure 3** (A) Principal coordinate analysis (PCoA) of *Oreochromis mossambicus* grouped by the farmed and wild populations; (B) PCoA of *Oreochromis mossambicus* individuals plotted using a genetic distance between individuals and color-coded by population.

sampling localities highlighted genetic distinctions. The genetic differentiation between farmed and wild *O. mossambicus* ranged from minimal (0.04 between the University of Zululand and Mhlathuze) to substantial (0.45 between uMphafa ponds and Matigulu) (Table S2). Importantly, pairwise $F_{ST}$-values indicated that the uMphafa ponds exhibited greater genetic similarity to the Fresca Fisheries Farm and Komati River sampling localities than to all sampling localities in KwaZulu-Natal. The probability estimates ($P$-values) for

**Table 4** Analysis of molecular variance (AMOVA) groupings at 14 microsatellite loci for *O. mossambicus* from four farmed and eight wild populations from Mpumalanga and KwaZulu-Natal.

| Source of variation | Sum of squares | Variance components | Percentage variation | *P* value <0.001 |
|---|---|---|---|---|
| Among groups (river or farm) | 400.09 | 1.09 | 21% | 0.000 |
| Among sites within groups | 915.25 | 1.46 | 27% | 0.000 |
| Within sites | 478.00 | 2.76 | 52% | 0.000 |
| **Total** | 1,793.34 | 5.31 | 100% | 0.000 |

comparisons between various farmed and wild sampling localities were predominantly significant, suggesting the presence of genetic distinctions (Table S2). Exceptions included comparisons between the Zini Fish Farm and two wild sampling localities (Mfolozi and Mhlathuze rivers), as well as between Fresca Fisheries Farm and the wild sampling localities from Komati and Mfolozi rivers, where no significant genetic differentiation was observed (Table S2). The *P*-values for genetic differentiation between farmed and wild sampling localities ranged from 0.001 for most farmed and wild sampling localities comparisons to 0.026 for Komati River and Loskop Dam sampling localities (Table S2).

## DISCUSSION

In this study, we investigated the genetic diversity and differentiation of farmed and wild *O. mossambicus* in KwaZulu-Natal and Mpumalanga provinces, South Africa, using 14 microsatellite loci. Samples from surrounding rivers were included to examine the origin of the farmed *O. mossambicus*. While the amplification of microsatellite loci was generally successful, resulting in some missing data in the matrix, the 14 microsatellite loci proved highly polymorphic, providing valuable insights into genetic diversity and population structure in both farmed and wild *O. mossambicus*. Specifically, the analysis revealed that farmed fish showed genetic similarity to nearby river populations, suggesting potential admixture. Genetic diversity was lower in farmed *O. mossambicus*, indicating limited admixture, while isolated natural populations exhibited distinct genetic characteristics.

### Genetic diversity
Genetic diversity indices indicated lower genetic diversity in farmed *O. mossambicus* from the uMphafa ponds compared to both other farmed sampling localities and their potential wild source counterparts. This assessment was based on overall genetic diversity indices calculated, including observed and expected heterozygosity. The diminished genetic diversity in the uMphafa ponds may be attributed to factors such as genetic drift, inbreeding, selection for captive conditions, imbalanced sex ratios of breeders through selection, and the effects of several generations of closed breeding (*Tave, 1993*; *Tave, 1999*; *Wang, 2002*; *Duong & Scribner, 2018*; *Lal et al., 2021*). Deviations from Hardy-Weinberg Equilibrium, potentially due to founder effects, non-random mating, and sampling methods were observed. This trend is common in cultured fish populations and aligns with findings from

previous studies on *Oreochromis* species and other aquaculture species (*Holtsmark et al., 2008*; *Simbine, Viana de Silva & Hilsdorf, 2014*; *Robledo et al., 2024*).

Genetic monitoring of farmed stock is imperative, and baseline data are crucial for guiding future population-specific conservation programs and research efforts (*Liu et al., 2018*). For the uMphafa ponds *O. mossambicus*, efforts should focus on maximising genetic variation and minimising inbreeding to maintain the heritability of valuable culture strains, especially if there is potential for these fish to be released or if they are already significant for breeding programs (*Lal et al., 2021*). Similarly to our findings, reduced genetic diversity in farmed populations, compared with wild ones, has been reported in various aquaculture studies (*Norris, Bradley & Cunningham, 1999*; *Pampoulie et al., 2006*; *Li, Wang & Bai, 2009*; *Briñez, Caraballo & Salazar, 2011*; *Aguiar et al., 2018*; *Ukenye & Megbowon, 2023*). The observed high genetic diversity in Zini Fish Farm and Fresca Fisheries farmed *O. mossambicus* may suggest effective genetic resource management. However, it is important to note that detailed information about the structure and composition of these farmed populations, such as the number of generations and broodstock management practices is not known. The diversity at Fresca Fisheries Farm could be due to broodstock sourced from different localities within the Komati River catchment. The *O. mossambicus* at Zini Fish Farm may have retained higher genetic diversity due to their access to natural breeding environments, which allows for more gene flow and mating opportunities with wild populations. This exposure to a broader genetic pool can enhance genetic variability. However, this situation also raises concerns about potential interactions with wild populations, including hybridisation and competition for resources. These findings suggest that differences in aquaculture practices and broodstock sourcing strategies can significantly influence the genetic makeup of cultured populations (*Boyd et al., 2005*; *Ying et al., 2018*; *Sonesson et al., 2023*). Further investigation into breeding strategies, generational turnover, and potential interactions with wild populations at these farms is needed to fully understand the drivers of the observed genetic diversity.

Overall, positive inbreeding values were observed for both farmed and wild *O. mossambicus* populations, except for the uMphafa ponds, which displayed a negative inbreeding value. Elevated inbreeding levels, a well-known consequence of culturing and widely reported in aquaculture species (*Briñez, Caraballo & Salazar, 2011*; *Aguiar et al., 2018*; *Geletu & Zhao, 2023*), can result from a limited number of founders and lead to inbreeding depression (*Leberg & Firmin, 2008*; *del Pazo et al., 2021*; *Patta et al., 2024*). The high inbreeding coefficients observed in wild *O. mossambicus* populations in this study could be attributed to habitat fragmentation and environmental degradation, which may have caused population isolation and limited gene flow (*Pavlova et al., 2017*; *Kim et al., 2023*). A recent comprehensive study on wild populations reported low genetic diversity and significant genetic differentiation among geographically distinct populations, further highlighting the impact of anthropogenic pressures on this species (*Mashaphu et al., 2024*). These findings emphasize the importance of managing both wild and farmed populations to mitigate further loss of genetic integrity and ensure their long-term viability.Additionally, population bottlenecks or a decline in effective population size could have reduced genetic diversity, further elevating inbreeding levels (*Naish et al., 2013*; *Allendorf, Hössjer*

& *Ryman, 2024*). However, the negative inbreeding coefficient at uMphafa ponds, while indicative of excess heterozygosity, did not align with the observed low genetic diversity. This discrepancy may be due to sampling bias (*Nei, 1972*) or the Wahlund effect, where the population could represent a mixture of subpopulations with distinct allele frequencies (*DeGiorgio & Rosenberg, 2009*). While maintaining inbreeding coefficients below 5% per generation is essential to prevent inbreeding depression, which can increase disease susceptibility, reduce fitness, and compromise environmental adaptation (*Tave, 1999*; *Geletu & Zhao, 2023*), further studies are needed to substantiate these claims and explore the genetic health of farmed *O. mossambicus* populations more thoroughly. This includes recognising the limitations of our current data and the necessity for additional research to confirm these patterns and their implications for the management of both farmed and wild populations in South Africa.

Conserving genetic variability in farmed fish populations involves strategies such as structured mating, where males and females are paired in ways that minimize genetic relatedness between mating pairs, thereby reducing inbreeding (*Fisch et al., 2015*; *Fernández, Villanueva & Toro, 2021*). This approach aligns with the optimal contribution strategy in breeding, which aims to maximize genetic diversity while maintaining desired traits (*Lind, Brummett & Ponzoni, 2012*; *May et al., 2024*). Additionally, restocking from wild populations or exchanging stock between farms could further counteract inbreeding depression (*Duong et al., 2017*; *Yusuf, Yisa & Sadiku, 2017*; *Liu et al., 2018*). For instance, introducing genetic stock exchanges between the University of Zululand, Zini Fish Farm, and uMphafa ponds *O. mossambicus* could enhance genetic diversity by introducing distinct stocks from each locality, while ensuring that geographic proximity is considered to avoid potential genetic disruptions. The genetic diversity observed in the Fresca Fisheries population suggests that this farm has maintained relatively high genetic variability in its *O. mossambicus* stock, as evidenced by the genetic diversity indices calculated in this study. This diversity is crucial for the continued viability of future generations, supporting adaptability and resilience to environmental changes. While this study did not assess specific farming practices, maintaining this genetic diversity through sound broodstock management is essential for ensuring the production of high-quality *O. mossambicus* seed for aquaculture. Further investigation into generational turnover and broodstock management practices would provide valuable context for understanding the genetic sustainability at Fresca Fisheries.

## Genetic population structure and genetic differentiation

The Bayesian STRUCTURE analysis identified nine distinct genetic clusters across the populations analysed, which included individuals from four farmed and eight wild *O. mossambicus* populations. The genetic composition revealed a high degree of genetic similarity between the farmed *O. mossambicus* from the University of Zululand and Zini Fish Farm and the wild *O. mossambicus* from the Mfolozi, Matigulu, and Mhlathuze rivers in KwaZulu-Natal. The clustering of these two farmed *O. mossambicus* implies a shared genetic origin, possibly sourced from the same rivers or interconnected water systems. This aligns with our understanding that the Zini Fish Farm is linked to the

Mlalazi estuary, connecting it to various river systems, including Mfolozi, Matigulu, Mhlathuze, Mvoti, and the Thukela rivers, emphasising the likelihood of genetic exchange between sampling localities. Although the exact origin of the University of Zululand's *O. mossambicus* stock is unknown, clustering analyses in this study indicate a potential genetic connection to wild *O. mossambicus* populations from the Mfolozi, Matigulu, and Mhlathuze Rivers. The high frequency of shared genotypes with the Mhlathuze River population suggests a possible genetic link. This could be due to shared source populations or historical gene flow between the farmed and wild populations in this region (*Aguiar et al., 2018*; *Fagbémi et al., 2021*). The uMphafa ponds *O. mossambicus* stands out as the most genetically distinct group compared to all other farmed and wild *O. mossambicus* analysed. This distinctiveness raises the possibility that it may not be a pure representation of the native *O. mossambicus* in the region and could potentially be a hybrid or derived from hybrids, possibly due to introductions of non-native *Oreochromis* species. In South Africa, hybridisation with introduced species is a known issue that complicates the genetic integrity of native *O. mossambicus* populations (*D'Amato et al., 2007*; *Firmat et al., 2013*; *Bills, 2019*). Notably, a small number of individuals sampled from the Thukela River share this distinct genetic profile with the uMphafa ponds, suggesting a possible historical connection. Further investigation, including a larger sample size from the Thukela River and analyses using additional genetic markers, is essential to evaluate the purity of the uMphafa ponds population and understand its potential relationship with the Thukela River *O. mossambicus*.

Similarly, the *O. mossambicus* at Fresca Fisheries Farm exhibited genetic distinctiveness from the surrounding wild *O. mossambicus*. However, some individuals from this farm clustered genetically with the wild *O. mossambicus* from the Komati River and Loskop Dam, suggesting that these wild sampling localities may serve as sources for Fresca Fisheries Farm. In contrast, the Pieter Vorster Dam *O. mossambicus* appeared highly genetically distinct, despite the belief that some farmed individuals at Fresca Fisheries Farm were sourced from this sampling locality. This indicates that the farmed *O. mossambicus* at Fresca Fisheries Farm does not accurately represent the genetic diversity of the Pieter Vorster Dam. The non-significant genetic differentiation further supports the notion that this farmed *O. mossambicus* originated from an admixture of *O. mossambicus* in the Komati River and Loskop Dam. However, this differentiation could also result from a founder event when establishing the farmed *O. mossambicus* and from broodstock input from other sources, which may be contributing to the observed high genetic diversity in this farm. Nei's genetic distance analysis in this study further supports the observed genetic patterns by highlighting the genetic similarities and differences between wild and farmed *O. mossambicus* populations. The small genetic distance between the University of Zululand and Zini Fish Farm populations suggests that these farmed populations may have originated from a common genetic source or have experienced historical gene flow. This observation aligns with previous studies that have indicated that small genetic distances can point toward potential source populations for cultured fish (*Macaranas et al., 1995*; *De Silva, 1997*; *Hall, 2001*; *Hassanien & Gilbey, 2005*; *Yusuf, Yisa & Sadiku, 2017*).

In the context of aquaculture, selective breeding emerges as a powerful tool for enhancing production and fostering a sustainable industry by cultivating strains with desirable traits (*Lind, Brummett & Ponzoni, 2012*; *Janssen et al., 2017*; *De Assis Lago et al., 2017*; *Robledo et al., 2018*; *Causey et al., 2019*). However, caution is imperative to prevent potential threats posed by selective breeding, such as the loss of genetic variation because of inbreeding. While this study did not focus directly on aquaculture practices or selective breeding and their impact on diversity, the recommendations by *Lind, Brummett & Ponzoni (2012)* for maintaining long-term genetic variation through selective breeding practices align with the importance of preserving high genetic diversity for sustainable aquaculture (*Lal et al., 2021*). The unique genetic characteristics identified in the Zini Fish Farm and Fresca Fisheries Farmed *O. mossambicus* in KwaZulu-Natal and Mpumalanga offer potential for future selective breeding efforts aimed at developing a local *O. mossambicus* strain. These sampling localities can potentially serve as valuable resources for supplementing broodstock in other *O. mossambicus* farms, contributing to increased genetic diversity in farmed stocks. However, it is important to note that uncontrolled mixing of genetically divergent individuals carries the risk of outbreeding depression, potentially leading to offspring with reduced fitness and adaptability (*Ward, 2002*; *Huff et al., 2011*; *Tsaparis et al., 2022*; *Vitt et al., 2023*). This underscores the necessity of careful management when considering gene flow between populations to maintain genetic health and long-term viability, particularly in the context of potential farm leakages that could contaminate wild populations (*Tsaparis et al., 2022*; *Vitt et al., 2023*). Additionally, using cultured fish for restocking or augmenting wild populations must be approached with caution. While aquaculture can significantly contribute to the sustainability of the industry by enhancing the genetic diversity of wild stocks, it also carries potential pitfalls. Genetic swamping, where the genetic integrity of wild populations is overwhelmed by the introduction of cultured fish, can lead to homogenisation and loss of local adaptations (*Ward, 2002*; *Vitt et al., 2023*). This process might reduce the overall fitness of the wild population and its ability to adapt to local environmental conditions. Moreover, the introduction of farmed individuals into the wild can result in the spread of diseases and parasites, further threatening the health of native populations (*Ward, 2002*; *Huff et al., 2011*; *Tsaparis et al., 2022*; *Vitt et al., 2023*). It is crucial to implement rigorous health screening protocols for farmed fish before their release into the wild to mitigate these risks. Overall, the enhancement of genetic diversity in farmed and wild *O. mossambicus* stocks through the use of cultured fish requires a balanced and carefully managed approach. Strategies should include the establishment of genetic management plans, regular monitoring of genetic health, and adherence to best practices in aquaculture to ensure the long-term sustainability and resilience of both farmed and wild populations.

## CONCLUSIONS

This study aimed to investigate the genetic diversity and differentiation of farmed *Oreochromis mossambicus* in KwaZulu-Natal and Mpumalanga, South Africa. Our findings revealed a complex genetic landscape, with some farmed populations closely resembling

their wild counterparts while others demonstrated distinct genetic profiles. Notably, the *O. mossambicus* from the uMphafa ponds in KwaZulu-Natal exhibit a unique genetic signature. Some individuals sampled from the Thukela River share this distinct profile, suggesting a potential historical connection between these two groups. This raises concerns about the source of the uMphafa populations, indicating they may have originated from the Thukela River or have hybridized with it. Given the known records of introduced *Oreochromis* species in South Africa, the uMphafa populations could potentially represent invasive species, emphasising the need for careful management to prevent genetic introgression between farmed and wild populations.

Conversely, the farmed *O. mossambicus* from the University of Zululand, Zini Fish Farm, and Fresca Fisheries Farm exhibit high genetic similarity to wild populations, suggesting their potential for restocking or supplementing wild populations, provided appropriate genetic safeguards are in place. Both Zini Fish Farm and Fresca Fisheries demonstrate high genetic diversity, making them valuable resources for enhancing farms with low genetic variation and for developing locally adapted strains suitable for sustainable tilapia farming in South Africa.

To ensure the long-term sustainability of *O. mossambicus* aquaculture and conservation efforts, it is crucial to quantify genetic diversity and population structure across South Africa. Future studies should prioritize evaluating the genetic integrity of both farmed and wild *O. mossambicus* populations before considering any supplementation into wild populations or the exchange and restocking of other *O. mossambicus* farms in South Africa. Advanced genetic techniques, such as single nucleotide polymorphism (SNP) analysis or an increased number of microsatellite loci, will provide a more comprehensive understanding of genetic variation. Additionally, increasing sample size and geographic coverage may enhance our assessment of population structure and genetic diversity across the country. By implementing effective management practices and conducting thorough genetic assessments, we can safeguard the genetic integrity of farmed and wild *O. mossambicus* populations while promoting sustainable aquaculture and conservation in South Africa.

## ACKNOWLEDGEMENTS

We thank the Ford Wildlife Foundation (ZA) for vehicle support and IdeaWild for equipment support. Special thanks to David Phiri, Lereko Tsoananyane, Ntaki Senoge, Angelica Kaiser, and Annelize Van der Merwe for their invaluable assistance in conducting surveys and data collection throughout Mpumalanga and KwaZulu-Natal. We also extend our gratitude to Mahomed Desai, Matthew Burnett, Emily Winter, Celine Hanzen, and Lance Holt for their help with DNA sample collection.

### Funding

This work was supported by the University of KwaZulu-Natal (ZA), the National Research Foundation (ZA, grant 98404), the South African Institute for Aquatic Biodiversity

(SAIAB), the Department of Forestry, Fisheries, and the Environment (DFFE), and the Agribusiness Development Agency (ADA). The funders had no role in study design, data collection and analysis, decision to publish, or preparation of the manuscript.

### Grant Disclosures
The following grant information was disclosed by the authors:
The University of KwaZulu-Natal (ZA).
The National Research Foundation: 98404.
The South African Institute for Aquatic Biodiversity (SAIAB).
The Department of Forestry, Fisheries, and the Environment (DFFE).
The Agribusiness Development Agency (ADA).

### Competing Interests
The authors declare there are no competing interests.

### Author Contributions
- Mahlatse Fortunate Mashaphu conceived and designed the experiments, performed the experiments, analyzed the data, prepared figures and/or tables, authored or reviewed drafts of the article, and approved the final draft.
- Gordon Craig O'Brien conceived and designed the experiments, prepared figures and/or tables, authored or reviewed drafts of the article, and approved the final draft.
- Colleen Thelma Downs conceived and designed the experiments, prepared figures and/or tables, authored or reviewed drafts of the article, and approved the final draft.
- Sandi Willows-Munro conceived and designed the experiments, analyzed the data, prepared figures and/or tables, authored or reviewed drafts of the article, and approved the final draft.

### Animal Ethics
The following information was supplied relating to ethical approvals (i.e., approving body and any reference numbers):
University of KwaZulu-Natal Animal Ethics Subcommittee (REF: AREC/023/020)

### Field Study Permissions
The following information was supplied relating to field study approvals (i.e., approving body and any reference numbers):
Permits for sampling were diligently obtained from Ezemvelo KwaZulu-Natal Wildlife (OP1583/2017, OP1432/2018, and OP871/2021) and Mpumalanga Tourism and Parks Agency (MPB. 56932).

### Data Availability
The datasets generated during the current study are available at Zenodo: Mashaphu, M. F., Downs, C. T., O'Brien, G. C., & Willows-Munro, S. (2024). Genetic assessment of farmed Oreochromis mossambicus populations in KwaZulu-Natal and Mpumalanga provinces, South Africa. https://doi.org/10.5281/zenodo.11085962.

## Supplemental Information

Supplemental information for this article can be found online at http://dx.doi.org/10.7717/peerj.18877#supplemental-information.

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
