# Peer review of "Genetic assessment of farmed Oreochromis mossambicus populations in South Africa"

_PeerJ, doi:10.7717/peerj.18877_

## Round 0.1 · original submission · Major Revisions

The manuscript presents significant insights, but major revisions are required to enhance its clarity and completeness. To improve the paper, it would be helpful to request that the authors provide more detailed explanations of their methodologies and statistical analyses. Despite the need for improvements in the presentation of the results and discussion section, the study and its outcomes are valid and may be relevant to the fields of conservation and aquaculture. Please, consider all the suggestions given by the reviewers in the new version of your manuscript.

·

Basic reporting

Basic Reporting
1. Clarity and Language:
The language is generally clear and professional. There are instances where phrasing could be clearer. For example, the sentence "Here the native Mozambique tilapia... although it is still not preferred over O. niloticus which is not native" (lines 23-26) could be rephrased for clarity. Some sentences are overly complex and could be simplified for better readability.
2. Introduction and Background:
The introduction provides a good context but could benefit from a more detailed explanation of include more details on the genetic issues associated with aquaculture and previous studies on Oreochromis mossambicus. This would help in understanding the knowledge gap the study aims to fill.
3. Literature Referencing:
The literature is generally well-referenced and relevant. However, there are some older references that could be updated to reflect the latest research.
4. Figures and Tables:
Figures and tables are relevant but should be reviewed for better quality and clarity. Figure 1 need to be clearer and more understandable. Ensure that all figures are of high quality.

Experimental design

Methodology:
• Can you provide more details on the software and specific parameters used for calculating genetic diversity indices and performing the STRUCTURE analysis?

• How were the microsatellite markers chosen, and are they consistent with those used in other similar studies?
Results
• The description of observed heterozygosity (Ho) and expected heterozygosity (He) is too unclear. The authors should provide more context about what these values imply for genetic health and diversity.

• Can you provide more context on what the observed heterozygosity values (0.55 to 0.65) mean in comparison to both wild populations and other farmed populations of O. mossambicus?
Presentation of Data:
• The figures themselves are somewhat unclear. For instance, the STRUCTURE analysis figures need better labeling and more descriptive legends to be fully understood.
Interpretation of Statistical Tests:
• The results mention significant genetic differentiation with a global FST value of 0.12, but there is insufficient explanation of what constitutes "moderate" differentiation in this context. Comparative data or references to similar studies are needed for better interpretation.
Discussion
• The statement about inbreeding depression becoming a significant issue is speculative without more concrete evidence. There should be a discussion of the potential limitations of the data and the need for further studies to confirm this.

Validity of the findings

1. Conclusions:
• The conclusions are generally well-stated and linked to the research question. However, they could be more concise and focused on the primary findings of the study.
2. Impact and Novelty:
• While the impact and novelty are not formally assessed, the study does contribute valuable information about the genetic diversity of farmed O. mossambicus. Highlighting the implications of these findings in the broader context of aquaculture practices would strengthen the discussion.

Additional comments

Summary
By addressing these points, the manuscript will be strengthened and better positioned for publication.

·

Basic reporting

The authors need to familiarize themselves with their results and I suggest that the authors should spend some more time reviewing more recent studies that have done similar work with the same type of markers even in different species. This will help them to coherently describe their results and discuss them better. Areas of improvement are provided in my additional comments.

Experimental design

it is fine but I suggest authors convert some of the tables to figures. This can easily be done as I have provided some guidelines. Areas of improvement are provided in my additional comments.

Validity of the findings

The results are valid, but I get a sense that the authors are over-stretching their results beyond the limits of their study, I can understand why they trying to do so but it should remain within the boundaries of the study. Areas of improvement are provided in my additional comments.

Additional comments

Line31: “The results indicated higher genetic diversity in the farmed populations compared with surrounding” Does this mean that farmed populations were more diverse than the wild populations? Please clarify.
Line33-35: “the uMphafa farmed population exhibited distinctive genetic characteristics, emphasising the need for careful monitoring and potential supplementation from other sources” How does a population having distinct genetic characteristics necessitate careful monitoring and supplementation of new genetics into the population? Please revise this and try to keep within the limits of your study when making general or specific recommendations.
Line35-38: “The study indicated the potential use of fish populations from select populations for breeding and broodstock supplementation, emphasising the importance of maintaining and closely monitoring genetic diversity in aquaculture practices.” I do not see the results that supports this conclusion from your analyses, please revise according to the limits of your study.
Line47: “long-term persistence of populations in the wild” what do you mean by this statement? Please revise.
Line49: “Wild populations are also an important genetic reservoir for aquaculture”, this sounds incomplete, if wild populations are reservoir for aquaculture, then? Please revise.
Line53:What do you mean by reproductive output? Did you mean reproductive fitness? Please revise, I guess you are trying to refer to the effects of inbreeding depression here.
Line56: Add “genus” after Oreochromis.
Line59: Remove “the” before “Nile tilapia”.
Line61: “(FAO 2002, FAO 2019)” Please review more literature.
Line62-63: “(Kocher et al. 1998)” Please cite more recent literature.
Line69: Instead of “genetic admixture” use hybridization as genetic admixture is generally used to refer to within-species genetic overlaps.
Line79: What natural resources that have you not mentioned? Please revise this.
Line78-79: “salinity and low temperature) of O. mossambicus is threatened by hybridisation (Moralee et al. 2000, DAmato et al. 2007).” Threatened by hybridisation with what?
Line84: Add “including” after “…as a case study”
Line111: What do you mean by “natural ponds”? is it earthen ponds?
Line123-143: Please mention the total number of fish used in the study and also mention the number of fish per site.
Line123-143: Were any morphometric measurements taken on the samples? If so please state here and present them.
Line146: Describe what you with the kit. Just stating the name of the used kit is not helpful to other scientists who might want to perform a study similar to yours.
Line149:Please include the NCBI Sequence ID or GenBank ID such as GU391020.2 for OM01 (https://www.ncbi.nlm.nih.gov/nucleotide/GU391020.2?report=genbank&log$=nuclalign&blast_rank=1&RID=AMJ5YBNB016)
Line157: Define A, D, B and C.
Line180-181: Revise this to “Tests for deviation from Hardy Weinberg equilibrium for markers were performed using GenAlEx”
Line181-182: Revise this to “Linkage disequilibrium (LD) between markers was estimated using Genepop version 4.7.0”
Line182-183: “In the latter analyses, chi-square p-values were obtained” what were the chi-square p-values for? Please be more elaborate describing what you did and why you did it.
Line191: Change “genetic divisions” to “sub-populations” or “ admixture”.
Line201-202: “Population structure was also visualised by plotting Neis genetic distances (Nei 1972) through Principal Coordinate Analysis (PCoA) in GenAlEx”. How did you convert genetic distances to principle components. Genetic distances are values between two individuals, while principal component coordinates for a single individual animal. Please revise this.
Line201-202: PCoA means Principal component analysis not “Principal Coordinate Analysis”, please review your methodology.
Line210: What do you mean by “Probability of FST-values”.
Line226-227: What did you do for markers/ loci that showed significant deviations from HWE.
Line229-230: Linkage disequilibrium is not determined for a single locus or marker but rather its determined between two Loci. Revise this.
Line233: I suggest you round off 4.98 to 5 as alleles occur in discrete numbers.
Line237-247: Your results indicate that at all loci the unbiased expected heterozygosity was higher than the observed. That seem to indicate that these populations actually inbreed, that is reflected in your Fis results across all loci, the values are mostly moderate to high (8/14, Fis range 26 – 074). Please revise the interpretation of your results, refer to material here (https://uwyo.edu/dbmcd/molmark/lect04/lect4.html#:~:text=If%20the%20observed%20heterozygosity%20is,of%20two%20previously%20isolated%20populations).
Line249-265: If something like alleles occur in discrete numbers in nature, please round the averages to off to the nearest whole number except the frequences.
Line250-255:Allele averages should be rounded off to the nearest whole number.
Line256-265: On average, Ho of your populations is lower than uHe and this is a sign of inbreeding. It is a bit strange that even your wild population show the same trend. This is also clearly reflected by the F in your table Table 3,your Fs are moderate to high except one population uMphapa ponds, such high Fs show high levels of inbreeding. Are you able to present the standard deviations or standard error of these F values. Please revise the interpretation of your results. (https://uwyo.edu/dbmcd/molmark/lect04/lect4.html#:~:text=If%20the%20observed%20heterozygosity%20is,of%20two%20previously%20isolated%20populations).
Line267-276: Which measure of diversity are you referring to here?
Line282: Use K instead of ∆K and also use “optimal sub-populations” instead of “optimal genetic portioning”.
Line281-294: Please revise this section in the context of the results, Structure results are not about alleles. I would suggest you refer to some papers that have done similar work like your and revise this. Example of such studies: https://onlinelibrary.wiley.com/doi/full/10.1002/aff2.197, https://link.springer.com/article/10.1186/s12862-020-1583-0.
Line296-305: PCoA is Principal component analysis not Principal coordinate analysis. You have two figures for Figure 3 (i.e. a and b) you do not refer to these in your description instead you give Figure 3. Secondly, could you clarify how you generated Figure 3a. Thirdly, label the axes, I guess that is the variation explained by the Principal components, eg Genetic variation explained by PC1, Genetic variation explained by PC2…
Line307 – 320: You can plot these distances in form of a tree using Splitstree (https://uni-tuebingen.de/en/fakultaeten/mathematisch-naturwissenschaftliche-fakultaet/fachbereiche/informatik/lehrstuehle/algorithms-in-bioinformatics/software/splitstree/)
Line329 – 338: You can plot these Fst values in a correlation plot (https://rkabacoff.github.io/datavis/Models.html)
Line348-349: Where are the results showing University of Zululand had lower genetic diversity than the natural populations? What do you mean by natural populations? Please use consistent terms.
Line349-351: Which results or parameters are you basing these claims? Please state the measure of genetic diversity you are referring to here.
Line352 – 353: I do not see the evidence that supports these claims. Actually most of your populations are inbreed except one population uMphafa.
Line357- 360: I still do seem to see where you get the evidence to this statement.
Line362-380: Not so different from the preceding paragraph, please internalize your results and combine the two paragraphs.
Line378 – 379: How does a population retain genetic diversity via having access to their natural environment for breeding.
Line391: I want to believe the 5% per generation. Please confirm this.
Line393 – 394: “potentially explaining the genetic differentiation observed between farmed and wild
populations.” This is not true, and I really do not see how this fits in your results.
Line384-389: Could explain why we see high inbreeding coefficients for wild populations in your study?
Line396: Structure mating where by males and females are mated in a design that ensure minimal relationships between mating pairs is a major strategy of maintaining low levels of inbreeding in farmed populations. Read a bit about optimal contribution strategy in breeding,
Line400-401: “Implementing good farming practices is essential for promoting adaptability, growth, low disease susceptibility, and increased fish production, benefiting the aquaculture industry.” This is out of place, please remove.
Line406 – 408: “The Fresca Fisheries population in Mpumalanga, indicating good genetics in the first generation, should be maintained for the viability of future generations, contributing to the production of high-quality O. mossambicus seed for aquaculture in Mpumalanga.” What evidence supports this from your study? Please keep in the limits of your study.
Line411-412: What do you mean structure analysis revealed 9 distinct genetic clusters encompassing four famed and 8 wild populations.
Line413: What do you mean by “close genetic affinity”?
Line421-424: Please revise this for better clarity.
Line424 – 428: “These findings are consistent with findings in other aquaculture studies that reported genetic similarities between farmed and wild populations of O. niloticus (Fagb¿mi et al. 2021), Cyprinus carpio (Napora-Rutkowski et al. 2017), Colossoma macropomum (Aguiar et al. 2018), and Brycon amazonicus (de Oliveira et al. 2018).” I do not find any connection of these studies or results with what you are reporting or discussing here. Please delete this and revise.
Line430: This distinctiveness implies that this population may not be representative of the wild populations. Which wild populations?
Line449 – 450: This is not clear, Nei’s genetic distances from your results? Please revise this.
Line463: Increasing selection limits is the same as employing large effective populations.
What do you think about the number of markers in your study? Don’t you think they are a limitation? How about the sample size, not another limitation?
Table: Add NCBI sequence ID of the microsatellite primers (you can use the blast tool from NCBI).
Table 3: I do not think its necessary to report PHWE.
Table 4: Present this data as a phylogenetic tree using this data.
Figure 1: Add number of samples besides the site name.
Figure 3: Define the percentages you have put in the y and x-axes. What do they mean?

Reviewer 3 ·

Basic reporting

The manuscript is well written and was an intereting read. The written English and technical editing was of a high standard.
The introduction was well constructed and provided a broad yet focused backdrop to the study.
I was not able to access the raw data (although these have been reposited) as the files were not accessible.
The necessary permits and ethical approvals are in place.
Extensive literature references were included (in some instances perhaps including too many to bring a point across). I have also indicated on the annotated copy some statements where the corresponding citations did not really sustantitate the corresponding text.
The tables and figures included adequately summarise the results and were correctly interpreted.

Experimental design

The research question is well defined to an extent. It comes out in the discussion that a "secondary" aim of the study might be to utilise the cultured animals to supplement the natural populations. Although this does not distract from the experimental setup it does somewhat confuse the subsequent discussion.
The experimental design follows the standards for population genetic analyses for comparing wild and cultured species. It is described in sufficient detail to allow for replication.
The methodology is sufficient to a large extent to address the research question. It would have benefitted from also including effective population size estimates as some of the conclusion revolves around maintaining effective population sizes in the cultured populations but as this was not estimated it is not clear what these sizes are in the cultured populations.
It would also have been pertinent to include more details on the breeding practices of the studied farms. What are they using as broodstock, how many generations of breeding have been going on. Are the broodstock supplemented (at the moment) or are advanced F generations being used. These all play into conclusions derived from this.
As a last note, measurements were taken of the sampled fish, but no mention made of what this was used for. Was it to ensure only adult fish were included in the study? If so, what is the minimum size requirement for this species to reach sexual maturity?

Validity of the findings

I do not doubt the validity of the majority of the findings but the article would have benefitted from further probing the populations like umPhafa which seem so distinct / genetically unique.
I am particularly concerned that considering the extent of potential hybridization in this species in Southern Africa, that no attempt was made to "verify" or test that the individuals included in the study were in fact O. mossambicus. A visual identification would not be enough to confirm this.

Annotated reviews are not available for download in order to protect the identity of reviewers who chose to remain anonymous.

---

## Round 0.2 · Minor Revisions

Thank you for submitting your manuscript for review.

Nevertheless, we have identified a recently published paper that bears a strong resemblance to yours: https://doi.org/10.1016/j.gecco.2024.e03043. We request that the authors provide an explanation regarding the distinctions between these two papers. Without such clarification, this submission may be considered a potential case of duplication.

·

Basic reporting

No comment

Experimental design

No comment

Validity of the findings

No comment

Additional comments

No comment

·

Basic reporting

The Manuscript should be fine for publication, however the authors have a very similar paper that has just been published recently https://doi.org/10.1016/j.gecco.2024.e03043. Could the authors provide clarification on the difference of these two papers, otherwise this seems like duplication.

Experimental design

The Manuscript should be fine for publication, however the authors have a very similar paper that has just been published recently https://doi.org/10.1016/j.gecco.2024.e03043. Could the authors provide clarification on the difference of these two papers, otherwise this seems like duplication.

Validity of the findings

The Manuscript should be fine for publication, however the authors have a very similar paper that has just been published recently https://doi.org/10.1016/j.gecco.2024.e03043. Could the authors provide clarification on the difference of these two papers, otherwise this seems like duplication.

---

## Round 0.3 · accepted · Accept

Dear Authors,
I am pleased to confirm that your paper has been accepted for publication in PeerJ.
Thank you for submitting your work to this journal.

With kind regards,